



# The version 3 OMI NO₂ standard product

Nickolay A. Krotkov[1], Lok N. Lamsal[2,1], Edward A. Celarier[2,1], William H. Swartz[3,1], Sergey V. Marchenko[4,1], Eric J. Bucsela[5], Ka Lok Chan[6], Mark Wenig[6]

[1]Atmospheric Chemistry and Dynamics Laboratory, Goddard Space Flight Center, Greenbelt, MD, US

[2] GESTAR, Universities Space Research Association, Columbia, Maryland, US

[3] Applied Physics Laboratory, Johns Hopkins University, Laurel, Maryland, US

[4] Science Systems and Applications, Inc., Lanham, Maryland, US

[5] SRI International, Menlo Park, California, US

[6] Department of Physics, Ludwig-Maximilians-University, Munich, Germany

*Correspondence to*: Nickolay A. Krotkov (Nickolay.A.Krotkov@nasa.gov)

**Abstract.** We describe the new, version 3.0 NASA Ozone Monitoring Instrument (OMI) standard nitrogen dioxide (NO₂) products (SPv3). The products and documentation are publicly available from the NASA Goddard Earth Sciences Data and Information Services Center (https://disc.gsfc.nasa.gov/uui/datasets/OMNO2_V003/summary). The major improvements include: (1) new spectral fitting algorithm for NO₂ slant column density (SCD) retrieval; (2) higher resolution ($1^{o}$ latitude and $1.25^{o}$ longitude) *a priori* NO₂ and temperature profiles from the Global Modeling Initiative (GMI) chemistry-transport model with yearly varying emissions to calculate air-mass factors (AMFs) required to convert SCDs into vertical column densities (VCDs). The new SCDs are systematically lower (by ~10%-40%) than previous, version 2, estimates. Most of this reduction in SCDs is propagated into stratospheric VCDs. Tropospheric NO₂ VCDs are also reduced over polluted areas, especially over Western Europe, the eastern US and eastern China. Initial evaluation over unpolluted areas has shown that the new SPv3 products agree better with independent satellite and ground based FTIR measurements. However, further evaluation of tropospheric VCDs is needed over polluted areas, where the increased spatial resolution and more refined AMF estimates may lead to better characterization of pollution hotspots.



## 1 Introduction

Emissions and concentrations of nitrogen oxides ($NO_x$=NO+$NO_2$) are regulated in several countries, as nitrogen dioxide ($NO_2$) is a toxic pollutant (US EPA, 2017) and $NO_x$ leads to the formation of surface-level ozone, acid rain and particular matter (Seinfeld and Pandis, 1998). $NO_x$ may also indirectly impact

climate through the formation of free tropospheric ozone (Jacob et al., 1996), a greenhouse gas, and secondary aerosols that scatter solar radiation and cool Earth's surface (Shindell et al., 2009). Major sources of $NO_x$ include fuel combustions, soil, and lightning.

Away from sources of tropospheric pollution, nearly 90% of the $NO_2$ total vertical column density is found in the stratosphere. There, it is approximately zonally symmetric and varies

meridionally with season. Stratospheric $NO_2$ is produced primarily by the oxidation of nitrous oxide ($N_2O$) transported from the troposphere. It catalytically destroys ozone and suppresses ozone loss by other catalytic mechanisms through the sequestration of active radical species (Seinfeld and Pandis, 1998).

$NO_2$ has strong spectral absorption lines in the visible (Vis) and near ultraviolet (UV) range,

which permit its measurement by remote sensing techniques. A new generation of spectroscopic ground-based instruments can measure total (Herman et al., 2009) and tropospheric (Hönninger et al., 2004; Spinei et al., 2014) $NO_2$ columns at high temporal resolution. The first space-based $NO_2$ observations started in the mid-90s with the Global Ozone Monitoring Experiment (GOME) instrument (Burrows et al., 1999; Martin et al., 2002; Richter et al., 2005). Similar measurements, but at higher

spatial resolution, continued with the SCanning IMaging spectrometer for Atmospheric CHartographY (SCIAMACHY: 2002–2012; Bovensmann et al., 1999), the Ozone Monitoring Instrument (OMI: 2004-present; Levelt et al., 2006), and GOME-2 (2006-present; Callies et al., 2000; Valks et al., 2011). Of these, OMI offers the highest spatial resolution, longest record, and lowest instrument degradation (Dobber et al., 2008; Marchenko and DeLand, 2014; Schenkeveld et al., 2017).

Satellite $NO_2$ data have been used as a proxy for 1) $NO_x$ emissions (van der A et al., 2017; Beirle et al., 2011; Boersma et al., 2015; Castellanos and Boersma, 2012; Curier et al., 2014; Ding et al., 2015; Duncan et al., 2014, 2016, de Foy et al., 2014, 2015, 2016; Ghude et al., 2013; Jaeglé et al., 2004; Konovalov et al., 2006, 2010; Lamsal et al., 2011; Liu et al., 2016; Lu et al., 2015; Lu and Streets,



2012; Martin et al., 2006; McLinden et al., 2016; Mijling and Van Der A, 2012; Richter et al., 2004, 2005; Russell et al., 2012; Stavrakou et al., 2008; Streets et al., 2013; Vinken et al., 2014; Zhang et al., 2007; Zhou et al., 2012), 2) $NO_2$ deposition (Nowlan et al., 2014), and 3) emissions of co-emitted gases, including other pollutants, like particulate matter, and greenhouse gases, such as $CO_2$ (Berezin et

al., 2013; Konovalov et al., 2016; Reuter et al., 2014).

There are two operational OMI $NO_2$ products: the NASA Standard Product (SP) (Bucsela et al., 2013) and the Dutch OMI $NO_2$ (DOMINO), produced by the Royal Netherlands Meteorological Institute, KNMI (Boersma et al., 2011). Both products use the Differential Optical Absorption Spectroscopy (DOAS) spectral fitting approach (Platt and Stutz 2006) to derive $NO_2$ slant column

density (SCD), which represents the total $NO_2$ amount [molecules $cm^{-2}$] along the average solar radiation path through the atmosphere as observed from OMI. After separation of tropospheric and stratospheric SCDs, these are converted to the respective $NO_2$ vertical column densities (VCDs) using model-derived air mass factors (AMFs): VCD = SCD/AMF. The previous NASA algorithm (version SPv2) used the same $NO_2$ SCDs as DOMINO v2 (Boersma et al., 2011), employing different

approaches to the stratosphere-troposphere separation (STS) and AMF calculation (Bucsela et al., 2013). Both products were in general agreement and produced similar regional trends in tropospheric VCDs (Krotkov et al., 2016), but comparison of OMI stratospheric $NO_2$ VCDs with other independent measurements revealed that they were overestimated by as much as 40% over unpolluted regions (Belmonte Rivas et al., 2014). The overestimation was traced to the common DOAS retrieval step (Van

Geffen et al., 2015; Marchenko et al., 2015).

This paper describes the new OMI operational $NO_2$ Standard Product, version 3 (SPv3), which is available from the NASA Goddard Earth Sciences Data and Information Services Center (GES DISC: https://disc.gsfc.nasa.gov/uui/datasets/OMNO2_V003/summary/). For version 3, we have developed a new DOAS spectral fitting algorithm, described in Section 3, that has brought OMI $NO_2$ SCDs into

much better agreement with independent satellite- and ground-based measurements and with model simulation results (Marchenko et al., 2015). Other changes include the use of higher spatial resolution *a priori* $NO_2$ profiles from the Global Modelling Initiative (GMI) chemistry and transport model, with updated, year-dependent emissions (Strode et al., 2015) and new, higher resolution temperature profiles





and tropopause height from the NASA MERRA model (Rienecker et al., 2011) discussed in section 2. Sections 4 and 5 compare the SPv3 with the previous version and with ground-based and satellite data.

## 2. Observations and model climatology

### 2.1 OMI measurements

The OMI instrument (Levelt et al., 2006) on the Earth Observing System Aura satellite (Schoeberl et al., 2006) is a pushbroom Ultraviolet – Visible (UV-Vis) spectrometer that measures the Earth's backscattered radiance and solar irradiance. The EOS-Aura satellite is flying in a sun-synchronous polar orbit with an equator crossing time of about 13:45 local time (ascending node). The swath width of OMI is 2600 km, enabling global daily coverage with a nadir field of view (FOV) size of 13 km x 24

km (along-track x across-track). OMI measurements have been radiometrically stable, as evidenced by regular evaluations of the instrument sensitivity changes (Dobber et al., 2008; Marchenko and DeLand, 2014; Schenkeveld et al., 2017). Comprehensive monitoring of the instrument's mission-long performance shows less than 3% degradation in radiances and irradiances in the 400-470 nm spectral range, stable long-term wavelength registration ($\Delta\lambda\sim0.002$ nm, with $\sim0.001$ nm seasonal fluctuations),

stable instrument slit function ($\sim0.1\%$) and stable stray light contamination in radiance and irradiance ($\sim0.5\%$ in the visible range; Schenkeveld et al., 2017). These qualities ensure generation of a consistent, long-term data record of $NO_2$ needed for the estimation of global trends, emissions, and other applications. Beginning in 2007, radiance measurements in some FOVs have been affected, apparently by a physical blockage of the entrance optics, rendering those measurements useless; this is called the

"row anomaly" (Dobber et al., 2008). Rejection of the anomalous FOVs leads to complete global coverage in two days instead of one, as before the row anomaly.

### 2.2 GMI model

Calculation of the AMF relies on an *a priori* $NO_2$ profile shape. The SPv3 AMF calculation uses the GMI 3-dimensional chemical transport model (CTM) simulation in the troposphere and stratosphere

(Duncan et al., 2007; Strahan et al., 2013). The GMI CTM uses a stratosphere-troposphere chemical mechanism, natural and anthropogenic emissions, and aerosol fields from the Goddard Chemistry





Aerosol Radiation and Transport (GOCART) model (Chin et al., 2014). It simulates tropospheric processes such as $NO_x$ production by lightning, scavenging, and wet and dry deposition. Meteorological fields, including temperature profile and troposphere pressure are the results of the Modern-Era Retrospective Analysis for Research and Applications (MERRA), and have 72 levels from the surface

to 0.01 hPa with a resolution ranging from ~150 m in the boundary layer to ~1 km in the upper troposphere and lower stratosphere. GMI simulations with MERRA have been evaluated in the troposphere and stratosphere. Strode et al. (2015) showed good agreement with tropospheric $O_3$ and $NO_x$ trends in the U.S. in a 1990–2013 hindcast simulation. Strahan et al. (2016) demonstrated realistic seasonal and interannual variability of Arctic composition using comparisons to Aura Microwave Limb

Sounder (MLS) $O_3$ and $N_2O$. We have found GMI's $NO_2$ simulation in both the troposphere (Lamsal et al., 2015) and stratosphere (Spinei et al., 2014; Marchenko et al., 2015) to be in good agreement with other independent measurements.

As in SPv2, the *a priori* profiles for SPv3 are monthly means of daily GMI profiles, sampled at the OMI overpass time (13:30-14:00 local time). The changes in the GMI simulation are summarized in

Table 1. Galactic cosmic rays (GCRs) were added to the model as an important source of stratospheric $NO_x$ at high latitudes.  The NO photodissociation rate, j(NO), was reduced by 40%, consistent with recent recommendations (M. Prather, personal communication, 2016). As NO photodissociation leads to loss of $NO_x$ in the stratosphere, reduction of j(NO) increases stratospheric $NO_2$ relative to the GMI simulation used in SPv2.

**3 Algorithm description**

As mentioned before, the SPv3 algorithm makes important improvements to the SPv2 approach, including a new OMI-optimized DOAS spectral fit to determine SCDs (S) and the improvement of AMFs for both the stratosphere and troposphere ($A_{strat}$ and $A_{trop}$). The stratosphere-troposphere separation (STS) algorithm remains unchanged from Bucsela et al. (2013). The main steps are depicted

in Figure 1 and described in more detail in the following subsections.



### 3.1 New SCD retrieval

In the new spectral fitting approach (Marchenko et al., 2015), we address certain shortcomings of the conventional DOAS approach, as applied to OMI retrievals. Conventional DOAS relies on very precise wavelength calibration, and simultaneously determines the trace gas absorptions and magnitude of the

inelastic rotational Raman (RR) scattering effect (Chance and Spurr, 1997; Grainger and Ring, 1962; Joiner et al., 1995). However, it is quite sensitive to the selection of the spectral fitting window, the order of the closure polynomial, and, most of all, to very slight misregistration between the radiance and irradiance wavelengths. We apply a multi-step, iterative – rather than simultaneous – retrieval procedure for all interfering species in the broad spectral window from 402 nm to 465 nm.

10       The statistical characteristics of the individual OMI solar irradiance measurements (Marchenko and DeLand 2015) compelled us to use monthly-averaged, rather than daily, spectra.

        In most spectral measurements, the RR effect imposes by far the largest signal in the spectral reflectances (radiance/irradiance). Our first step is to use the spectral structure of the RR signal to (1) ascertain and correct the wavelength offset between radiance and irradiance (~0.002nm; *cf.* the 0.21 nm

spectral sampling step), and (2) remove the RR signal prior to estimating the SCDs. We assess the wavelength dependence of the shifts by splitting the entire fitting window into multiple overlapping micro-window segments and evaluating the RR spectrum amplitudes and wavelength adjustments for each segment. To account for the RR line-filling patterns, we use a linear combination of the atmospheric (Joiner et al., 1995) and the liquid-water (Vasilkov, 2002) RR spectra, convolved with the

wavelength- and cross-track-dependent OMI spectral transfer function (Dobber et al., 2006).

        Other steps in the algorithm include the estimation of, and correction for, spectral under-sampling patterns (Chance et al., 2005) and aggressive suppression of instrumental noise.

### 3.2  AMF calculation

The method of AMF calculation remains the same as in SPv2 (Bucsela et al., 2013), which agrees well

with independent estimates (Lorente et al., 2016). To calculate stratospheric and tropospheric AMFs we use a pre-computed dimensionless scattering weight vector **W**, (also known as the Box-AMF; Platt and Stutz 2006). **W** describes the relationship between S for a column (stratospheric or tropospheric) and



the local VCD, $V_i$, in each atmospheric layer i within the column (Palmer et al., 2001; Martin et al., 2002):

$$S = \sum_i W_i \times V_i = A \times \sum_i V_i = A \times V \tag{1}$$

$W$ is pre-computed using the radiative transfer program TOMRAD (Dave, 1965), accounting for multiple molecular (Rayleigh) scattering in an atmosphere bounded by a Lambertian surface. Since the Lambertian equivalent surface reflectance (LER) is assumed to be wavelength independent, $W$ varies smoothly with wavelength (within ~20%) across the $NO_2$ fitting window. Therefore, we calculate a

single $W$ vector, representative of the entire spectral fitting window, which is stored in a look-up-table (Bucsela et al., 2013). Stratospheric and tropospheric AMFs are calculated, separated at the climatological MERRA monthly tropopause pressure (i.e., $A_{trop}$ and $A_{strat}$ in Fig. 1). In the stratosphere, $W$ is approximately constant with altitude and is determined by the solar and viewing zenith angles: $W_{i,strat} \approx \sec(SZA) + \sec(VZA)$. In the free troposphere, $W_{i,trop}$ increases with altitude and strongly

depends on the cloud radiance fraction and optical centroid pressure (Sneep et al., 2008; Stammes et al., 2008; Vasilkov et al., 2009). In the boundary layer and under cloud-free conditions, $W$ depends most strongly on altitude and surface pressure and reflectance (Vasilkov et al., 2017).

The AMF for a stratospheric or tropospheric column is computed as the vertical integral of the $NO_2$ profile-shape weighted average of $W$ (Eq. 1) using the *a priori* profiles described in Section 2.2.

These profiles capture the interannual (Lamsal et al., 2015) and seasonal variability (Lamsal et al., 2010) of the AMF. The SPv3 uses yearly varying monthly mean $NO_2$ profiles from 2004 to 2014. For dates starting in 2015, the 2014 monthly profiles are used. The $W$ is corrected for the monthly mean GMI temperature profile as described in Bucsela et al. (2013), since the S retrieval algorithm relies on a constant temperature (220K) $NO_2$ cross-sections.

OMI $NO_2$ column averaging kernels (AKs) can be calculated from the scattering weights ($W$) and corresponding AMFs for stratospheric or tropospheric columns: $AK = dV/d\,V_i = W/A$ (Eskes and Boersma, 2003). The $AK$s are used in data assimilation; observational system simulation experiments (OSSEs), and in comparisons with vertically resolved measurements and CTM models.



### 3.3 Stratosphere-Troposphere separation (STS).

The STS algorithm remains the same as in the previous version (Bucsela et al., 2013), which shows overall good agreement with the independent STRatospheric Estimation Algorithm from Mainz (STREAM) – a verification algorithm for the Sentinel 5 Precursor Tropospheric Monitoring Instrument

(TROPOMI) STS (Beirle et al., 2016). The $V_{strat}$ and $V_{trop}$ are retrieved separately under the assumption that the two are largely independent (Fig. 1). The stratospheric field is computed first, beginning with creation of a gridded global initial field $V_{init} = S/A_{strat}$, assembled from data taken within ± 7 orbits of the target orbit. An *a priori* estimate of the tropospheric contribution to this field, $S_{trop\_ap}/A_{strat}$, based on a monthly GMI model climatology and OMI cloud measurements, is subtracted, and the potentially

contaminated grid cells where this contribution exceeds $0.3 \times 10^{15}$ molec. cm$^{-2}$ are masked. A three-step (interpolation, filtering, and smoothing) algorithm (Bucsela et al. 2013) is then applied to fill in the masked regions and data gaps and to remove residual tropospheric contamination. The resulting stratospheric vertical column field $V_{strat}$ is converted to a slant column field using $A_{strat}$ and subtracted from the measured S to provide $S_{trop}$, leading to the desired $V_{trop} = S_{trop}/A_{trop}$ (Fig. 1). The $S_{trop}$ can be

combined with independently calculated $A_{trop}$ to develop customized regional $V_{trop}$ products, for example, at higher spatial and/or time resolution (Kuhlmann et al., 2015; Laughner et al., 2016; Lin et al., 2014; Russell et al., 2011, 2012).

### 3.4 Retrieval noise and bias

We compare noise and biases in SPv2 and SPv3 by analyzing retrievals over homogeneous non-

polluted Pacific regions with negligible tropospheric contribution (Fig. 2). The data are filtered to minimize geophysical, observational, and cloud-induced variability. The selection criteria result in low SCDs with the largest DOAS fitting uncertainties and should be treated as upper bounds on uncertainties over non-polluted mostly cloud-free regions (Table 2). In this relatively clean region, uncertainties in the AMF and STS are much smaller than in polluted regions, where (1) the tropospheric

column is much larger than the stratospheric column, and (2) the STS algorithm is filling in where data were masked (Beirle et al., 2016; Bucsela et al., 2013).



Our new OMI DOAS spectral fitting algorithm (Marchenko et al., 2015) greatly reduces the positive biases (i.e., constant offset in S $\sim +1.2 \times 10^{15}$ molec. cm$^{-2}$ and multiplicative factor 0.1*S) reported in the version 2 retrievals (Van Geffen et al., 2015), albeit with a slightly increased noise (0.9 ± 0.3 $\times 10^{15}$ molec. cm$^{-2}$, Table 2). We estimate the noise as a standard deviation of the mostly cloud-free

S retrievals over nearly homogeneous Pacific regions. The upper limit corresponds to the tropical regions and near-nadir observations, while the lower limit corresponds to large solar and/or OMI zenith angles (i.e., large S). The noise increased ~20% with time: from ~0.8 ± 0.3 $\times 10^{15}$ molec. cm$^{-2}$ in 2005 to ~1.0 ± 0.3 $\times 10^{15}$ molec. cm$^{-2}$ in 2015.

Figure 2 compares probability distribution functions (PDFs) of retrieved $V_{strat}$ and $V_{trop}$ derived

by both versions over the equatorial South Pacific region for four months in 2011. As expected, the known overestimation in $V_{strat}$ is reduced by a constant offset ~0.6 $\times 10^{15}$ molec. cm$^{-2}$ in the new retrievals, bringing them into closer agreement with independent satellite (Adams et al., 2016; Belmonte Rivas et al., 2014; Marchenko et al., 2015) and ground-based FTIR measurements (Section 5). The noise in $V_{strat}$, estimated as monthly standard deviation, is unchanged from the previous version ~0.10 ±

0.04 $\times 10^{15}$ molec. cm$^{-2}$ (Table 2). It is much lower than the upper bound estimate of the noise in $V_{init}$ = S/2 ~0.45 ± 0.15 $\times 10^{15}$ molec. cm$^{-2}$, which is a result of the smoothing step in the STS algorithm (Bucsela et al., 2013).

The noise in $V_{trop}$ ~0.45 ± 0.04 $\times 10^{15}$ molec. cm$^{-2}$ (Table 2) is estimated using its monthly standard deviation (Fig. 2). It is consistent with the upper bound of the noise in $V_{init}$ = S/$A_{strat}$ assuming

near nadir observations and $A_{strat}$~2. The deviation of the mean $V_{trop}$ from $V_{trop-ap}$ is less than 0.1 $\times 10^{15}$ molec. cm$^{-2}$, as is expected given how the STS algorithm works (Bucsela et al., 2013).

Over polluted regions the "bias" in $V_{trop}$ is poorly defined, as (1) it may be larger and more variable (Fig. 3) due to the larger spatial-temporal variability in tropospheric VCDs, (2) the $A_{trop}$, being computed from model-based monthly mean profiles, may not accurately represent the true AMF

(Lorente et al., 2017), and (3) the STS procedure fills in the stratospheric field over polluted regions using measurements from some distance away  (Beirle et al., 2016; Bucsela et al., 2013).

The noise can be reduced with time averaging, e.g. creating monthly, seasonal, and annual average $V_{trop}$. Pixel averaging techniques, such as oversampling and pixel rotation along wind direction,


have been developed to increase effective spatial resolution and signal-to-noise, leading to improved detection and characterizations of the point emission sources (Fioletov et al., 2015; de Foy et al., 2015; Kuhlmann et al., 2014; Lu et al., 2015; McLinden et al., 2016).

## 4   Comparison with previous version

Figure 3 shows global monthly mean $V_{strat}$ and $V_{trop}$ difference maps from the previous version 2.1 for December 2006, when we see the largest differences between the versions. The SPv3 $V_{strat}$ (SPv3) is uniformly reduced by 0.5-0.8 $\times 10^{15}$ molec. cm$^{-2}$. One notices very large reductions in $V_{trop}$ (~2-5 $\times 10^{15}$ molec. cm$^{-2}$) over heavily polluted regions in Europe, the eastern US and, particularly, eastern China. However, for exceedingly large $V_{trop} > 10^{16}$ molec. cm$^{-2}$ the relative difference between the two versions

is usually less than ~20%. The $V_{trop}$ reductions are caused by combined effects of smaller SCD (Fig. 3, top right) and changes in the updated emissions and spatial resolution of the *a priori* $NO_2$ profile shapes (Fig 3., bottom right). All these changes reduce $V_{trop}$ over most polluted areas of the world. By capturing the year-to-year changes in $NO_2$ profile shapes (Fig. 4), the updated emissions used in the new GMI simulation substantially change the $NO_2$ vertical distribution in the highly polluted regions,

lending more confidence in the observed rapid changes in $NO_x$ emissions around the globe in the last ten years (Krotkov et al., 2016). These changes reflect a considerable decline in $NO_x$ emissions between 2005 and 2011 over the US and western Europe, and an increase over China. The observed difference in $NO_2$ profiles between the two simulations could also arise from the changes in model resolution.

### 4.1 Impact on regional trends

Regional $V_{trop}$ maps and trends comparing OMI $NO_2$ from SPv2 and SPv3 are shown in Figs. 5-8. Figure 5 shows annual average $V_{trop}$ in 2005, 2010, and 2015 over eastern US for both versions as well as their differences. We see reductions up to ~2 $\times 10^{15}$ molec. cm$^{-2}$ over mostly polluted megacity regions in eastern US along major interstate I-95 from Baltimore to New York (I-95 corridor, red box in Fig. 5). Elsewhere, the reductions are less than $10^{15}$ molec. cm$^{-2}$, including major industrial regions with

coal-burning power plants in southwest Pennsylvania and Ohio River Valley (blue box in Fig. 5).





A signature of the change in model resolution can be seen in the difference map as subtle box-like artefacts. The significant $NO_2$ reduction with time is also evident. The reduction is a result of emission regulations on power plants and vehicles (Duncan et al., 2013; de Foy et al., 2015; Lamsal et al., 2015; Lu et al., 2015; Russell et al., 2012; Tong et al., 2015).

Figure 6 compares relative changes in $V_{trop}$ in 2005-2015 for the I-95 and Ohio regions calculated from the two retrievals and other polluted region discussed later. The relative trends are largely the same using both versions. $NO_2$ concentrations over polluted regions in the eastern US fell by more than 40%, as result of the Clean Air Act Amendments and follow-up regulations (Krotkov et al., 2016).

Figure 7 compares annual mean tropospheric $NO_2$ over western Europe in 2005, 2010 and 2015. One may notice large differences in $V_{trop}$ ~2-3 $\times 10^{15}$ molec. cm$^{-2}$ over densely populated and industrialized regions in southwest Netherlands, northwest Belgium, Westphalia in Germany (Randstad-Ruhr, blue box in Fig. 7), and along the industrial Po River valley in the northern Italy (red box in Fig. 7). The changes are much smaller ($<10^{15}$ molec. cm$^{-2}$) over less polluted regions. During the OMI

mission we see significant $NO_2$ reductions with time (~ 25% for Randstad-Ruhr and ~40% for the Po River valley) related to national regulations and EU air quality directives aimed at reducing emissions from transportation and power sectors and creating a sustainable living environment (Boersma et al., 2015; Castellanos and Boersma, 2012). As seen in the I-95 and Ohio Valley samples, SPv2 and SPv3 retrieved tropospheric columns give trends that are well within statistical uncertainties of each other for

both European regions (Fig. 6).

Figure 8 compares annual mean $V_{trop}$ over eastern China in 2005, 2010 and 2015. The maximum $V_{trop}$ values in pollution hot spots were reduced in new version, but areas with increased $V_{trop}$ can also be seen over Yangtze and Pearl River deltas. The $NO_2$ plumes over the coastal regions reach much farther offshore. In densely populated areas the plumes seem to spread farther into the suburban regions.

This could be the result of the increase in spatial resolution of the *a priori* profiles on the AMF calculation: in the lower panel, a signature of the much coarser grid ($2° \times 2.5°$) used in SPv2 can easily be seen.



The blue box in Fig. 8 outlines the region of the North China plane (NCP, blue box in Fig. 8), which has the world's largest $NO_2$ pollution, with the annual average $V_{trop} > 10^{16}$ molec. $cm^{-2}$. This is a result of the high density of coal-fired power plants, other industries, as well as dense traffic. The impact of the new version on $NO_2$ relative trends is more evident for the NCP than from the other

regions considered. Figure 6 shows that over the NCP the $NO_2$ peaked in 2010-2011 but decreased from the peak by ~50% by 2015 (Krotkov et al., 2016). The reduction is likely due to government regulations, economic slowdown, and technological improvements in limiting $NO_x$ emissions by vehicles, industry, and power generation (de Foy et al., 2016). The new version shows a 10-20% smaller increase in peak $NO_2$ in 2010-2013 but negligible changes in early and recent years (Fig. 6).

**4.2 Impact on lightning $NO_x$ emissions estimate**

Lightning produced $NO_x$ ($LNO_x$) plays an important role in tropospheric chemistry. Recent research has shown that satellite measurements are a useful tool for estimating $LNO_x$ (Boersma et al., 2005; Beirle et al., 2010; Bucsela et al., 2010; Pickering et al., 2016). Pickering *et al.* (2016) combined OMI $V_{trop}$ data with data from the World Wide Lightning Location Network (WWLLN) (Dowden et al., 2002; Lay *et*

*al.,* 2004*;* Virts *et al.,* 2013) to estimate the production efficiency (PE) of $LNO_x$ (moles per flash). Using SPv2 and WWLLN data from the Gulf of Mexico over 5 Northern Hemisphere summers (2007 – 2011), they obtained a mean PE value of $80 \pm 45$ mol/flash. Applying the same algorithm to SPv3 data, we obtain $77 \pm 45$ mol/flash; the difference with the SPv2 result is not statistically significant. Using the new SPv3 data will likely have little effect on $LNO_x$ PE estimates derived in other regions.

**5. Comparisons with independent measurements**

**5.1 Comparison with FTIR measurements in Tenerife**

Figure 9 shows an improved agreement of $V_{total}$ ($=V_{strat} + V_{trop}$) from SPv2 to SPv3 when evaluated against ground-based Fourier transform infrared (FTIR) spectrometer measurements at Izana, Tenerife (28.3°N, 16.5°W; Schneider *et al.*, 2005). Izana was chosen as the best candidate station in the Network

for the Detection of Atmospheric Composition Change (NDACC), whose data are publicly available (http://www.ndacc.org). It is a low-to-middle latitude site, remote from pollution sources, making $V_{total}$



measurements throughout the day (not just at sunrise/sunset), and has a long data record. The FTIR measurements made before, near, and after the OMI overpass time (all solar zenith angles <75°) were selected and corrected to the OMI measurement time. The seasonal mean differences with OMI SPv2 ranged from 25% to 35%, with the OMI $V_{total}$ always larger than the FTIR values. With SPv3, the mean

differences are reduced to ~10%, with OMI still higher, on average. We use the difference, ~0.3 $\times 10^{15}$ molec. cm$^{-2}$, as an estimate of the bias in $V_{strat}$ over non-polluted, low-latitude areas (Table 2).

### 5.2 Comparison with MAX-DOAS measurements in Hong Kong

In previous studies, $V_{trop}$ measured by OMI were seen to be systematically lower than MAX-DOAS measurements in highly polluted "hot spots" in urban environments (Chan et al., 2012; Wenig et al.,

2008). We have conducted a comparison with ground-based MAX-DOAS (tropospheric) $NO_2$ column measurements in the heavily polluted Hong Kong area to quantify the differences brought by the new version. The results are presented in Figure 10. In agreement with previous studies, monthly-averaged OMI data are systematically lower than the monthly-averaged ground based measurements, but are very similar for SPv2 and SPv3. The winter values are slightly higher in the new version, bringing them

closer to the MAX-DOAS data (*cf.* the bottom row of panels Fig. 8). Some reasons for the discrepancies between satellite- and ground-based $NO_2$ retrievals include the spatial averaging inherent in the large OMI field of view, the still quite coarse sampling of the *a priori* profiles and surface reflectance used for the AMF calculation, and the influence of aerosols, which have not been explicitly included in the AMF calculation. OMI shows similar annual variability as the MAX-DOAS data and the changes made

to the retrieval of the new $NO_2$ standard product does not significantly change the annual patterns.

### 5.3 Comparison with independent satellite retrievals

Figure 11 shows comparisons of OMI $V_{total}$ and $V_{strat}$ with independent satellite $NO_2$ data from the Global Ozone Monitoring Experiment-2 (GOME-2) (Pieter Valks, personal communication) and SCIAMACHY (Bovensmann et al., 1999) nadir measurements using the German Aerospace Center

(DLR) retrievals (version 5.02) over the Pacific region for March in 2005 and 2010. The OMI data were filtered so that only FOVs unaffected by OMI's so-called row anomaly (Dobber et al., 2008) were used.



The data were additionally filtered so only FOVs with a measured cloud radiance fraction of less than 0.5 were included. The Pacific region was chosen because it is relatively free of tropospheric pollution. Thus, virtually all the $NO_2$ column is in the stratosphere. Because stratospheric $NO_2$ increases largely monotonically during the day, as photochemistry repartitions nitrogen oxides (e.g., Bracher *et*

*al.*, 2005), observations made at different local solar times cannot be compared directly. Stratospheric $NO_2$ increases during the day from the time of the GOME-2 and SCIAMACHY overpasses (morning) to that of OMI (early afternoon), so the GOME-2 and SCIAMACHY data shown in Figure 11 have been adjusted to 13:45 local time, based on the diurnal variation of $NO_2$ simulated by the GMI CTM. Previous version retrievals exceed both SCIAMACHY and GOME-2 by 20–30%. The new SPv3

data are in much better agreement with the other satellite measurements, to within about 10%, except at higher latitudes, above 50°N. These comparisons are in general agreement with the ground-based FTIR measurements in Izana (Fig. 9). The observed difference at high latitudes could arise from the difference in retrieval algorithms, instrumental behavior, or imperfect photochemical correction.

Figure 12 shows comparisons of OMI SPv3 with GOME-2 separately for stratospheric and

tropospheric VCDs. Overall, $V_{strat}$ retrievals show better agreement, mostly well within the specified 0.5 $\times 10^{15}$ molec. $cm^{-2}$ uncertainty. However, over polluted regions in Eastern China and South Africa, OMI $V_{trop}$ fall below the GOME-2 values by 1-2 $\times 10^{15}$ molec. $cm^{-2}$. Although the retrieval algorithms for OMI and GOME-2 use a similar approach, the details of the retrievals differ quite greatly.

## 6   Conclusions

For the past 12 years, OMI has been making hyperspectral earthshine radiance measurements, including in the range 400-470 nm, where $NO_2$ has a strong, structured absorption feature that lends itself well to the DOAS retrieval technique. We have recently released a new, Version 3 OMI $NO_2$ Standard Product (SPv3) based on significant improvements in both the estimation of the $NO_2$ SCDs and the estimation of the AMFs. While the revised SCD estimates come from a new retrieval algorithm, the AMF refinements

relate to updates in the GMI chemical and transport model inputs, including revised photochemical rate constants, more extensive emission inventories, and a horizontal resolution that is twice as fine in both latitude and longitude.



The quantities of greatest interest are the tropospheric, stratospheric, and total VCDs. Here we provide the uncertainties in these VCDs and evaluate the changes in the VCDs from the previous version (SPv2), also showing the improved agreement between the SPv3 VCDs and independently measured values from ground- and space-based instruments.

5        In non-polluted areas $V_{trop}$ has not changed appreciably form SPv2 to SPv3. In more polluted areas, the $V_{trop}$ values have greatly decreased, from SPv2 to SPv3. Figure 3 shows that most of the decrease in the highly polluted areas is due to the change in SCD, with some additional decrease due to the changed AMF. With the currently adopted AMF estimates we anticipate the overall reduction in the OMI derived top-down anthropogenic $NO_x$ emissions and surface concentrations. Despite large absolute

differences, the relative temporal regional changes in $V_{trop}$ as well as estimates of lightning $NO_x$ production efficiency in free troposphere are not significantly affected in the revised data. Additional long-term ground-based column $NO_2$ measurements and surface concentration network data will be very helpful in validating the presented version 3 of the standard OMI $NO_2$ product.

**7  Data availability**

OMI $NO_2$ data used in this study have been publicly released as part of the Aura OMI standard $NO_2$ Product (OMNO2.003, DOI:10.5067/Aura/OMI/DATA2017) and can be obtained free of charge from the NASA Goddard Earth Sciences (GES) Data and Information Services Center new public web site: https://disc.gsfc.nasa.gov/uui/datasets/OMNO2_V003/summary. New OMI $NO_2$ overpasses as well as

daily and monthly maps are available from the NASA Aura validation web site: http://avdc.gsfc.nasa.gov/. The FTIR data at Izana as part of the Network for the Detection of Atmospheric Composition Change (NDACC) and publicly available (see http://www.ndacc.org and the Aura Validation Data Center, http://avdc.gsfc.nasa.gov).

**Competing interests**

The authors declare that they have no conflict of interest.





## Acknowledgments

We acknowledge the NASA Earth Science Division, specifically the Aura science team program, for funding OMI $NO_2$ product development and analysis. The Dutch-Finnish-built OMI instrument is part of the NASA EOS Aura satellite payload. KNMI and the Netherlands Space Agency (NSO) manage the
OMI project. The authors thank Susan E. Strahan for helpful discussions of the GMI model simulations. We thank Thomas Blumenstock (Karlsruhe Institute of Technology) for help using of the Izana FTIR data.

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



**Table 1:** GMI model specifications used in SP $NO_2$ retrieval

| Model parameter | SPv2.1 (released 2012) | SPv3 (released 2016) |
|---|---|---|
| Spatial resolution (lat x lon) | 2°x2.5° | 1°x1.25° |
| Meteorological fields | GEOS5.1 | MERRA |
| Fossil Fuel $NO_x$ emissions | Constant 2005–7 | Time-dependent |
| Biomass Burning $NO_x$ emissions | Constant 2005–7 | Time-dependent |
| Lightning $NO_x$ coefficients | Calculated from 2005-2007 of older simulation | Calculated from over 20 years of MERRA reanalysis |
| Tropospheric aerosols | Constant year 2001 GOCART | Time-dependent GOCART |
| Stratospheric aerosols | Constant year 2000 | Time-dependent (IGAC) |
| Galactic cosmic rays | No | Yes |
| j(NO) scaling factor | 1.0 | 0.6 |



**Table 2:** SP $NO_2$ retrieval biases and noise estimated over non-polluted mostly cloud-free (Cloud Radiance Fraction < 0.3) Pacific Ocean regions in July 2011 [ $\times 10^{15}$ molec. $cm^{-2}$ ].

| Parameter | SPv2.1 (released 2012) | SPv3.0 (released 2016) |
|---|---|---|
| Bias in S | $max(1.2, 0.1 \times S)$ [1] | $\pm 0.5$ [2] |
| Noise in S | $0.8 \pm 0.2$ [3] | $0.9 \pm 0.3$ [4] |
| Bias in $V_{init} = S/A_{strat}$ [5] | $+ 0.60$ | $\pm 0.25$ |
| Noise in $V_{init} = S/A_{strat}$ [5] | $0.40 \pm 0.10$ | $0.45 \pm 0.15$ |
| Bias in $V_{strat}$ | $+ 0.6$ [6] | $+ 0.3$ [7] |
| Noise in $V_{strat}$ [8] | $0.10 \pm 0.04$ | $0.10 \pm 0.03$ |
| Bias in $V_{trop}$ [9] | $\pm 0.1$ | $- 0.1$ |
| Noise in $V_{trop}$ [10] | $0.36 \pm 0.03$ | $0.45 \pm 0.04$ |

1) Estimated as constant off-set value ~1.2 (Van Geffen et al., 2015) for $S < 12 \times 10^{15}$ molec. $cm^{-2}$ and

multiplicative value ~0.1×S for S >$12 \times 10^{15}$ molec. $cm^{-2}$ (Marchenko et al., 2015) ;

2) Intercomparison of independent DOAS fitting algorithms (Van Geffen et al., 2015; Zara et al., 2016);

3-4) Mission time average value of standard deviation in S over Pacific regions in 2011; upper limit corresponds to small S. The noise increased by ~20% during OMI mission: from ~$0.8 \times 10^{15}$ molec. $cm^{-2}$ in 2005 to ~$1.0 \times 10^{15}$ molec. $cm^{-2}$ in 2016 (Zara et al., 2016);

5) Upper limit of uncertainty in $V_{int}$ is estimated from uncertainties in S assuming $A_{strat} \sim 2$

6) Relative to satellite limb observations (Belmonte Rivas et al., 2014)

7) Comparisons with independent satellite and ground based FTIR measurements at Izana;

8) Estimated as the standard deviation of $V_{strat}$ over the tropical South Pacific region (5S to 15S and 120W to 160W) in 2011. Uncertainty reflects noise seasonal dependence (Fig. 2).

9) Estimated as the difference between mean OMI retrieved and a priori = $<V_{trop}>$ - $<V_{trop\_ap}>$ over unpolluted homogeneous tropical south Pacific region;

10) Estimated as the standard deviation of $V_{trop}$ over the tropical South Pacific region (5S to 15S and 120W to 160W) in 2011. Uncertainty reflects noise seasonal dependence (Fig. 2).

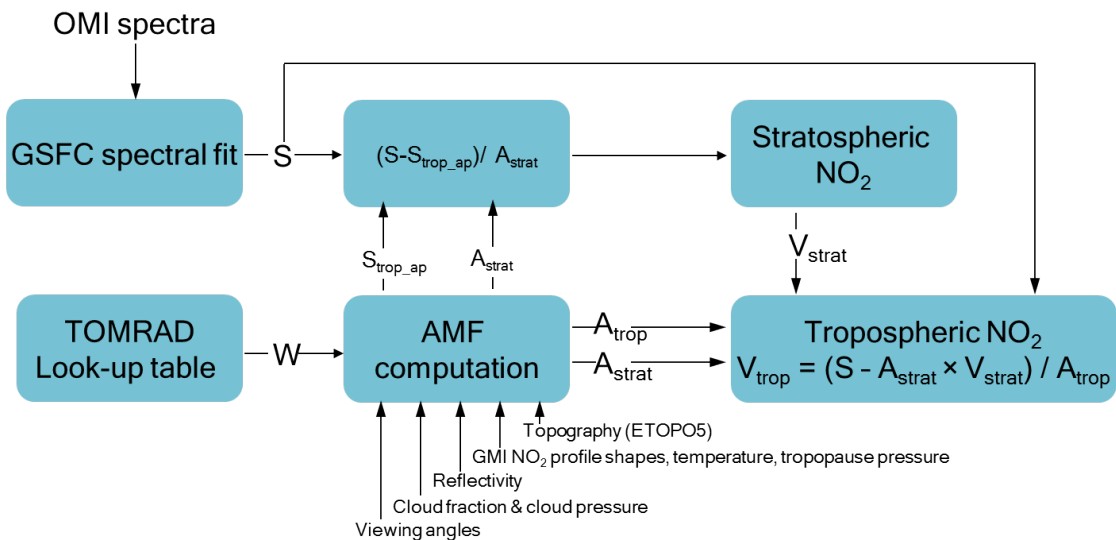

**Figure 1: Schematic description of the OMI NO₂ processing algorithm. S variables represent slant column densities (SCD); A - air mass factors (AMF). V variables represent vertical column densities (VCD). W denotes the scattering weight (Eq. 1), pre-computed using the radiative transfer program TOMRAD.**



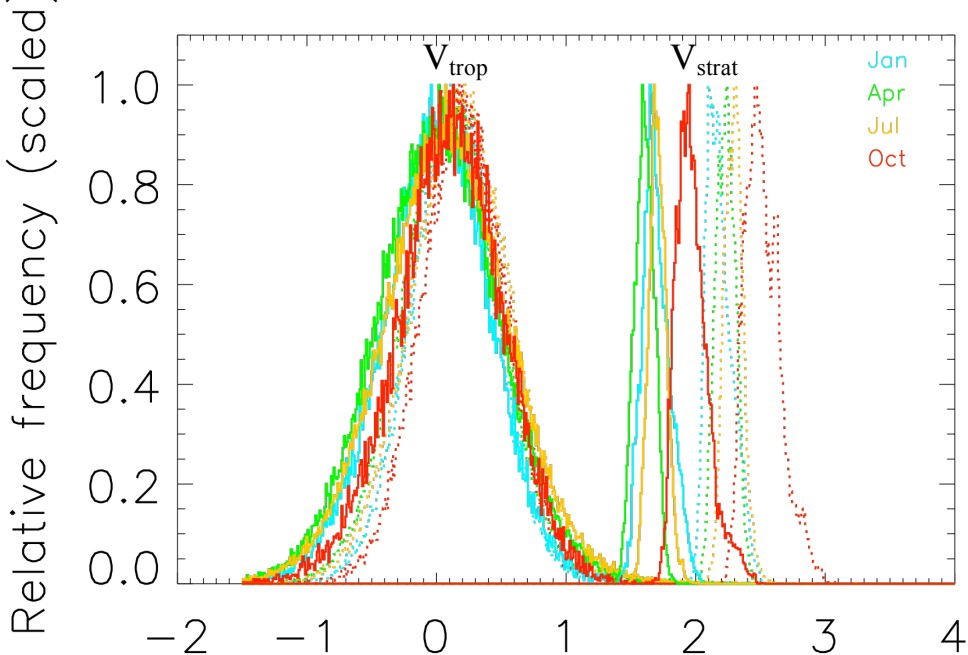

**Figure 2. Probability distribution functions (PDF) of the new SPv3 (solid lines) and previous version SPv2 (dashed lines) VCDs [×10$^{15}$ molec. cm$^{-2}$] retrieved in the Pacific region 15S < lat < 5S and 160E < lon < 130W, during 2011. The width of the V$_{trop}$ is**
5   **used as proxy for estimated noise in V$_{trop}$ ∼ 0.5 ×10$^{15}$ molec. cm$^{-2}$ (Table 2).**





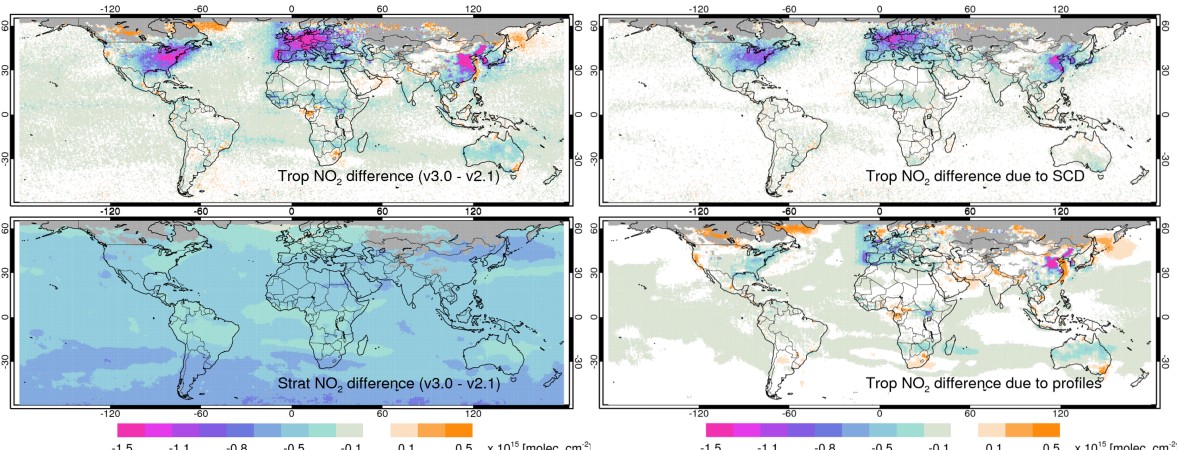

**Figure 3. OMI NO$_2$ SPv3 – SPv2.1 difference maps for December 2006: tropospheric VCD (top left), stratospheric VCD (bottom left). Change in V$_{trop}$ due to new SCD (top right), and change in V$_{trop}$ due to new a-priori NO$_2$ profile shapes (bottom – right).**



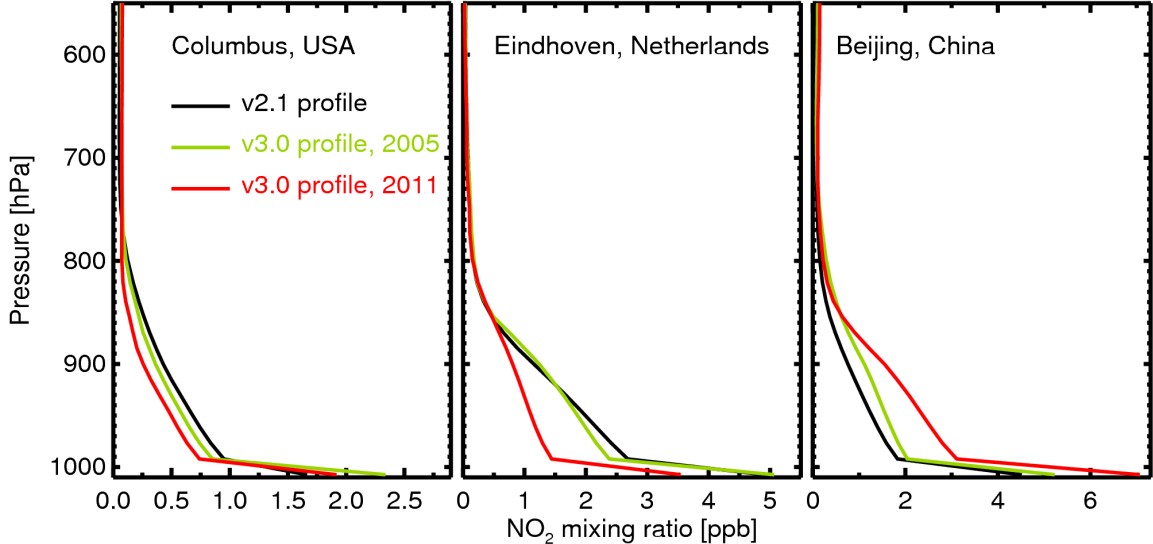

**Figure 4** Monthly average vertical distribution of NO₂ in July from GMI over selected locations in eastern US, western Europe, and China. The color lines show the average NO₂ profiles derived from the new high resolution (1°×1.25°) GMI simulation for 2005 (green) and 2011 (red). The black line shows NO₂ profiles derived from previous (SPv2) GMI simulation at 2°×2.5°.





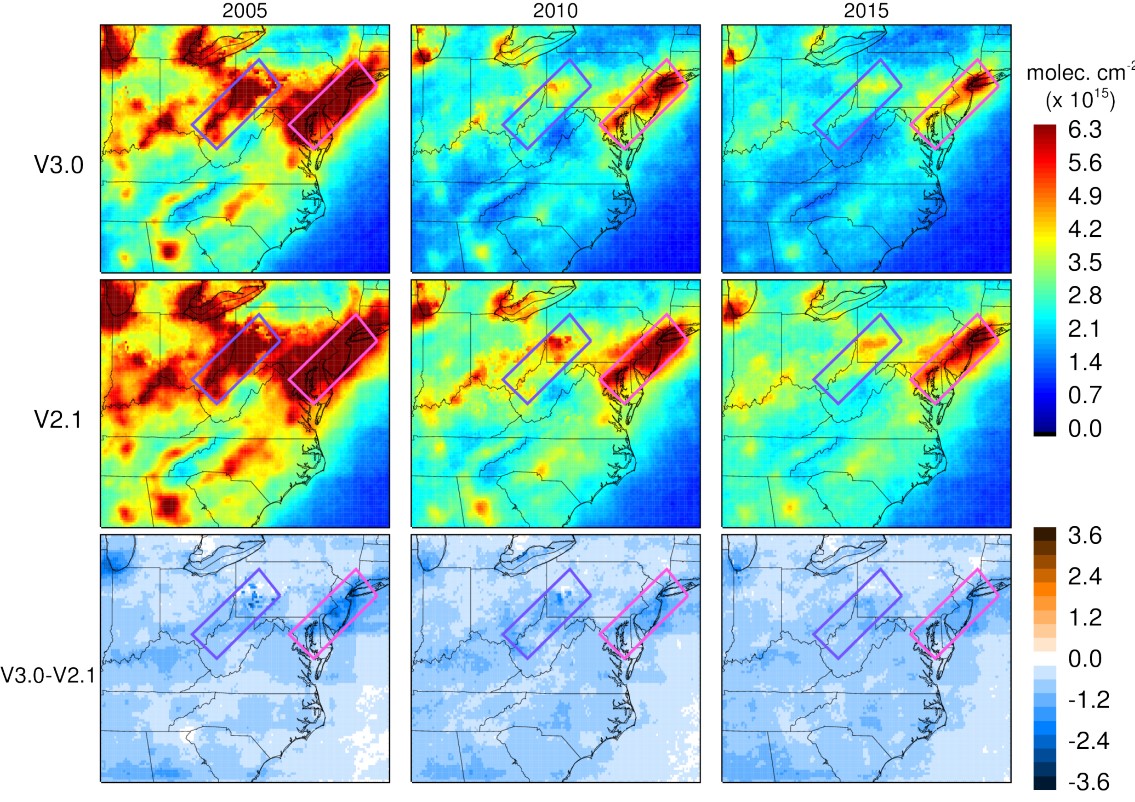

**Figure 5. Annual average OMI NO$_2$ V$_{trop}$ maps over eastern US for 2005, 2010, and 2015: SPv3 (top), SPv2 (middle) and the difference: SPv3 – SPv2 (bottom). The blue box outlines the Ohio River valley and southwestern Pennsylvania region with the predominant emissions from coal-fired power plants (Ohio in Fig. 6). The red box outlines the megalopolis from Washington, DC to New York along the I-95 interstate highway (I-95 corridor in Fig. 6) with predominant emissions from mobile sources. The regions have been discussed in Krotkov et al. (2016).**





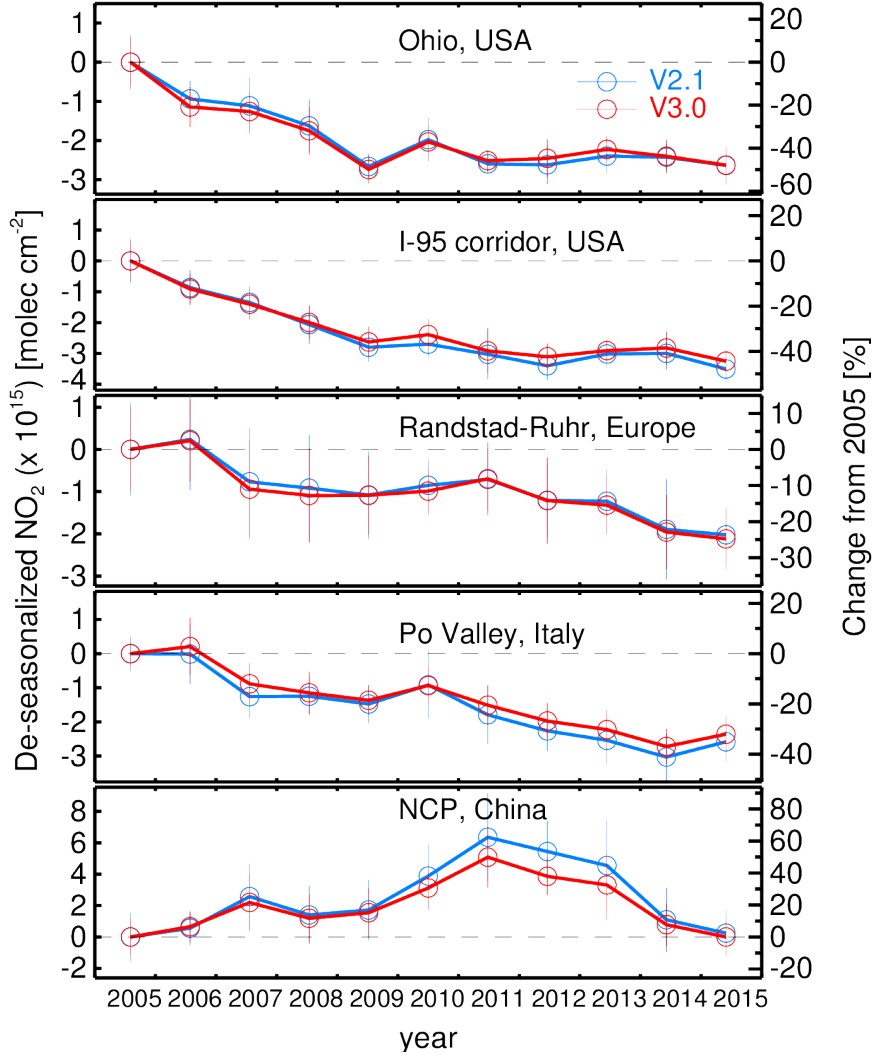

**Figure 6. Annual average OMI NO₂ V_trop regional trends for selected regions outlined in Figs. 5, 7-8. The regions in eastern US and eastern China have been presented in Krotkov et al. (2016).**



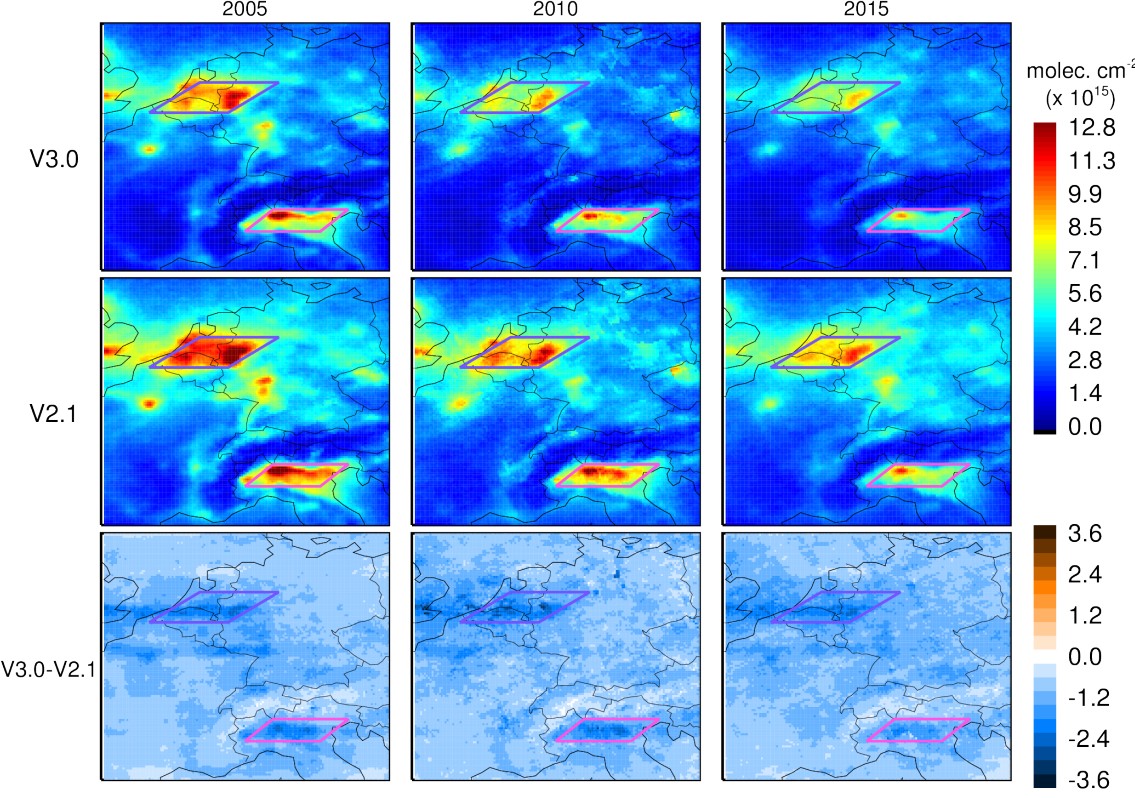

**Figure 7. OMI SP V3 (top) and V2 (middle) and difference V$_{trop}$ maps over western Europe for 2005, 2010, and 2015. The boxes outline the densely populated and industrialized regions in southwest Netherlands, northwest Belgium, and Westphalia in Germany (blue box: Randstad-Ruhr in Fig. 6), and in industrial Po River valley in the northern Italy (red box: Po Valley in Fig. 6).**



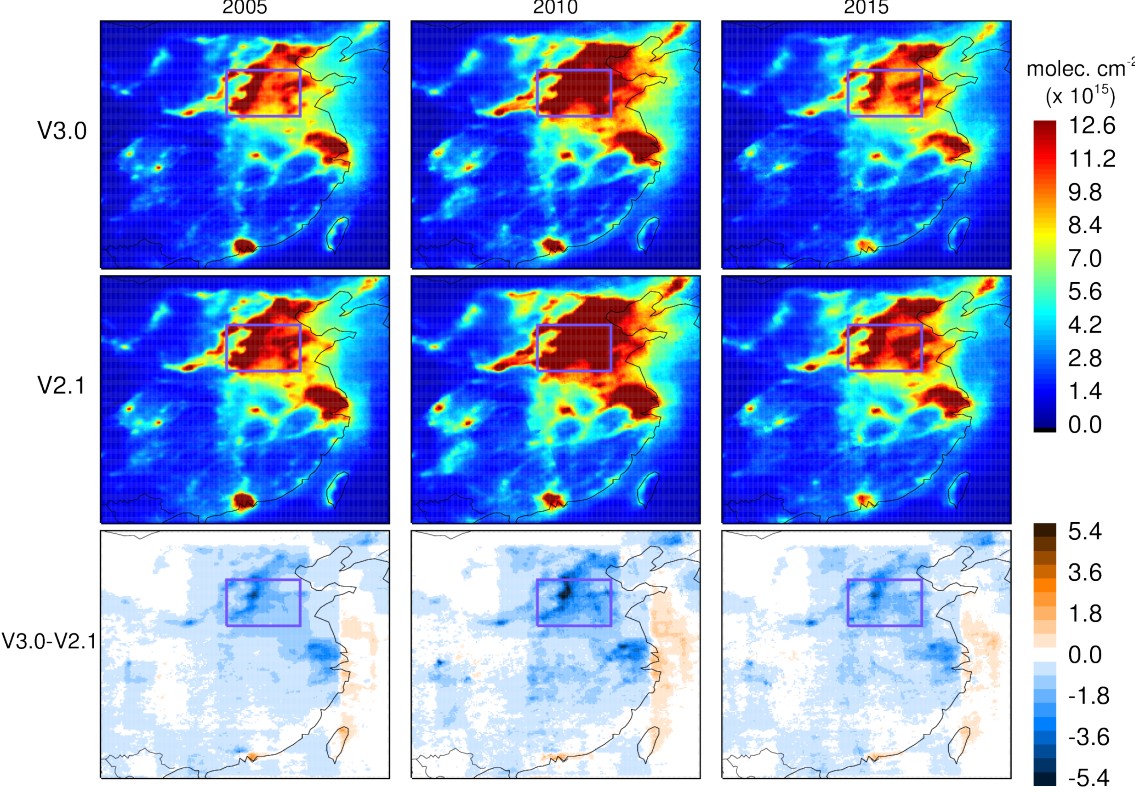

**Figure 8. OMI SPv3 (top) and SPv2.1 (middle) and difference V$_{trop}$ maps over eastern China for 2005, 2010, and 2015. The box outlines the densely populated and industrialized region in North China Plane (NCP in Fig. 6). The region has been discussed in Krotkov et al. (2016).**




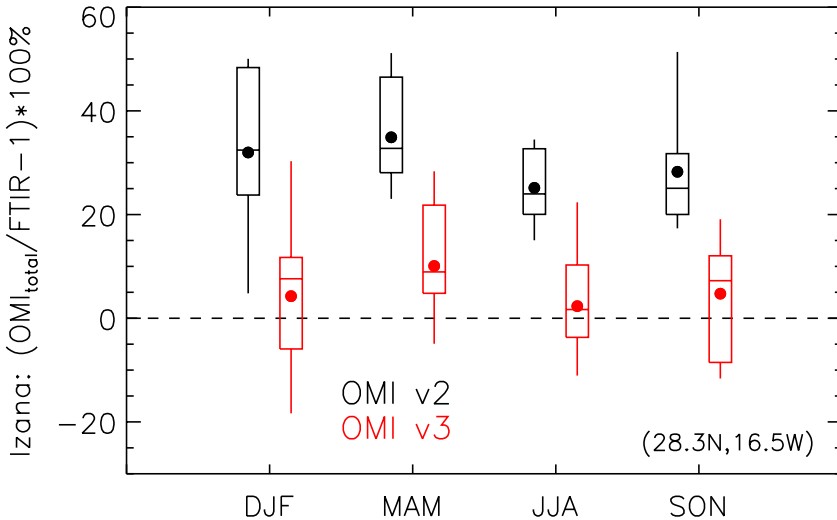

**Figure 9. OMI $V_{total}$ versus ground-based FTIR at Izana in Tenerife (28.3°N, 16.5°W), seasonally for 2005–2011. SPv2 and SPv3 are shown for FOVs within 50 km of the ground-based site. Photochemical corrections have been made for the OMI overpass time. Box-and-whisker plots show 10, 25, 50, 75, 90 percentiles; the dots in middle are the means.**





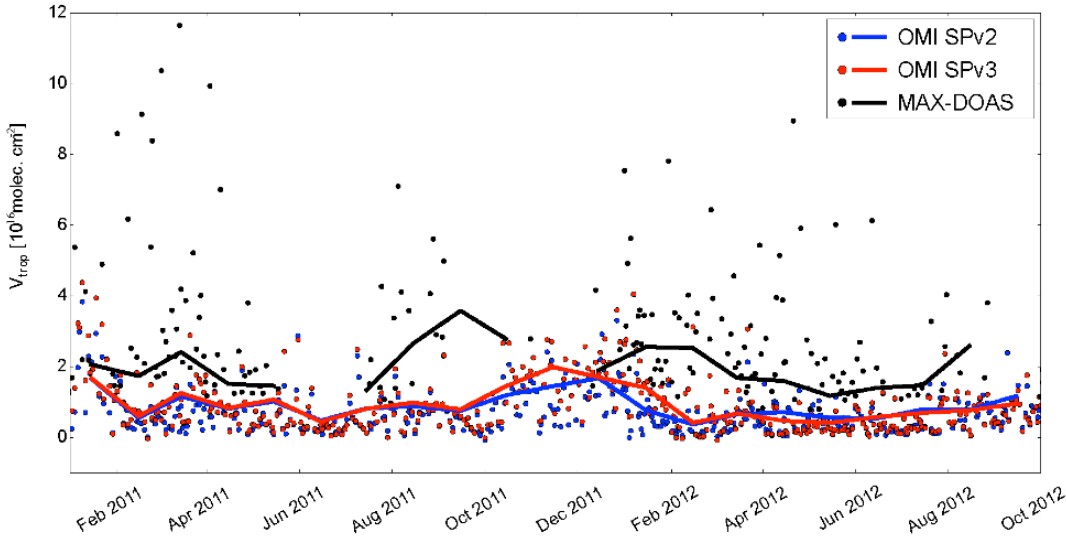

**Figure 10.** Comparison of OMI data with MAX-DOAS data retrieved in Hong Kong. The OMI data has been interpolated and gridded on a 1km x 1km grid and then the pixel for the measurement site has been extracted. The dots show daily values and the lines monthly averages. Note that compared to other parts of eastern China, $V_{trop}$ values do not decrease significantly and even increase for some months, probably because of the improved a-priori profiles better capturing the sharp contrast between clean ocean profiles and steep vertical gradients in one of the most densely populated cities in the world.





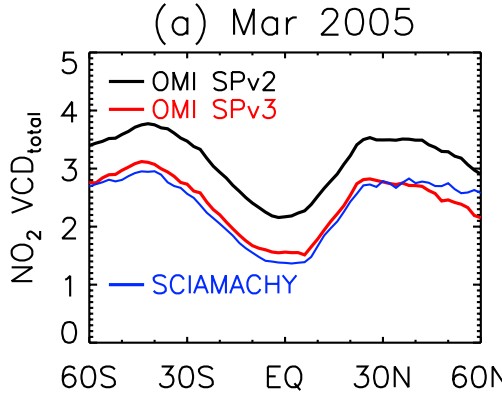 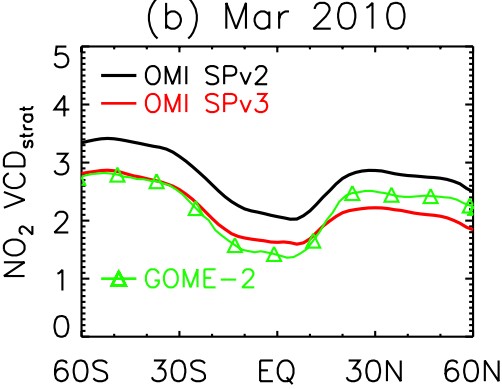

**Figure 11. OMI, SCIAMACHY and GOME-2 retrievals over the Pacific Ocean region (180°W–140°W) for (a) VCD$_{total}$ in March 2005 and (b) VCD$_{strat}$ in 2010. The SCIAMACHY and GOME-2 data have been adjusted to the local time of the OMI overpass by making photochemical corrections based on the diurnal variation simulated by the GMI CTM**





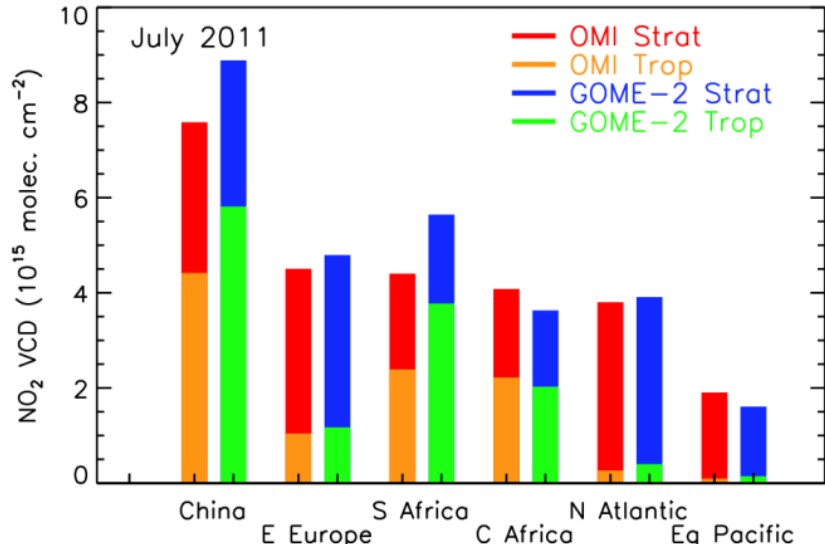

**Figure 12. Mean tropospheric and stratospheric NO$_2$ VCDs retrieved using new version SPv3 over several polluted and un-polluted regions: China: 110-125E, 30-42N, East Europe: 33-48E, 42-50N, South Africa: 25-35E, 22-30S, Central Africa: 10-30E, 0-14S, North Atlantic: 25-35W, 45-51N, Equatorial Pacific: 150-160W, 5S-5N. In all cases, the GOME-2 data have been adjusted to the local time of the OMI overpass by making a photochemical correction to the stratospheric portion of the total column, based on the diurnal variation simulated by the GMI CTM.**