# Peer review of "The version 3 OMI NO2 standard product"

_Atmospheric Measurement Techniques, 2017_

## Referee Comment (RC1) · Anonymous Referee #3 · 27 Apr 2017

This is a review of the manuscript "The version 3 OMI NO2 standard product" by Krotkov et al. Overall, this is a very well written, straightforward manuscript that clearly describes updates to the Goddard OMI NO2 retrieval. The overall reductions in tropospheric column NO2 that are evident in the new retrieval have important implications for those who study air quality and policy makers that use these results to guide policy decisions. I recommend publication of this manuscript and offer some minor comments as follows.

General comments: How do the GMI model results used for the a priori compare to other models such as GEOS-Chem? Prior studies (e.g., Huang et al. 2015) show some differences between GMI and GEOS-Chem with the latter model comparing better to in-situ observations above the boundary layer. This may impact the results shown in this study and should be discussed.

What is the justification of reducing J(NO)? This is very significant. While Prather is

held in high regard, it is important to offer some further explanation for this. Will this result be published soon?

For Figure 3, it would be very useful to not only show the differences in the retrievals but to also show the retrievals themselves for both the stratosphere and the troposphere.

How are negative values treated in the new retrieval? Are there fewer negative values in SP3?

Specific comments:

Figure 10. Please add error bars to the figure. It's difficult to interpret this figure otherwise.

Technical Corrections:

Page 9, line 23 Fig.3 should be Figure 3. If this is the first time it is referenced.

Page 12, line 1: I believe "North China Plane" should be "North China Plain", the caption of Figure 8. should be modified accordingly

Huang, J., Liu, H., Crawford, J. H., Chan, C., Considine, D. B., Zhang, Y., Zheng, X., Zhao, C., Thouret, V., Oltmans, S. J., Liu, S. C., Jones, D. B. A., Steenrod, S. D., and Damon, M. R.: Origin of springtime ozone enhancements in the lower troposphere over Beijing: in situ measurements and model analysis, Atmos. Chem. Phys., 15, 5161-5179, doi:10.5194/acp-15-5161-2015, 2015.

---

## Referee Comment (RC2) · Anonymous Referee #4 · 14 Jun 2017

The manuscript describes the new V3 algorithms for OMI NO2 retrievals. The new algorithm includes significant improvements compared to the previous algorithms, for example, the new spectral fitting algorithm for NO2 slant column density (SCD) retrieval and the higher resolution a priori NO2 and temperature profiles. The manuscript is clear and well written but I think it requires additional comparison with ground-based observations. Therefore, I recommend publication after addressing the following comments:

1) P6 L10-11 How much do you expect this choice of the monthly spectra to affect the correct representation of the day-to-day variability?

2) Fig. 3 I think it would be useful to have the same plots for June or July in order to have an idea of the difference at northern high latitudes, which are uncovered in

December.

3) Fig. 6 This large difference over NCP, China made me think about the impact this change in the algorithm could have on emission estimation, in particular in China where recently significant emission reduction has been probed from space-based observations. Can you estimate or at least speculate on how top-down emission estimates might be affected by these algorithm changes, as compared to some of the conclusions provided in existing literature?

4a) Section 5.2 In a previous work (Ialongo et al., 2016) where V2.1 and V3 total columns are compared to Pandora observations in Helsinki (Finland), the difference between V3 and V2.1 retrievals are quite systematic, with V3 sensibly smaller than Pandora (as compared to V2.1). Can you speculate about the difference between your results with this previous paper?

Ialongo, I., Herman, J., Krotkov, N., Lamsal, L., Boersma, K. F., Hovila, J., and Tamminen, J.: Comparison of OMI NO2 observations and their seasonal and weekly cycles with ground-based measurements in Helsinki, Atmos. Meas. Tech., 9, 5203-5212, doi:10.5194/amt-9-5203-2016, 2016.

4b) In general, I think the manuscript could get stronger with slightly more comparison to independent ground-based observations, similarly to what was presented by (Lamsal et al., 2014). While the main scope of this manuscript might not be the comprehensive validation of the V3 product (this could be addressed in a separate paper, perhaps), I would suggest including one or two more pictures, for example something similar to fig. 6 and 7 in Lamsal et al. (2014), including for example Pandora measurements or additional max-doas stations. According to this previous paper, such ground-based data should be available to the authors.

Lamsal, L. N., Krotkov, N. A., Celarier, E. A., Swartz, W. H., Pickering, K. E., Bucsela, E. J., Gleason, J. F., Martin, R. V., Philip, S., Irie, H., Cede, A., Herman, J., Weinheimer, A., Szykman, J. J., and Knepp, T. N.: Evaluation of OMI operational standard NO2

column retrievals using in situ and surface-based NO2 observations, Atmos. Chem. Phys., 14, 11587–11609, doi:10.5194/acp- 14-11587-2014, 2014.

5) Vasilkov et al. (2017) discussed the effect of the varying observation geometry on the NO2 vertical column retrieval. Their findings suggest that the NO2 vertical columns would typically increase (at least in this test orbit over the American continent), when taking into account a geometry-dependent Lambertian equivalent reflectivity (LER) in the NO2 retrieval algorithm. Can you comment in the manuscript on how this could affect your retrievals?

6) Section 5.3 P14 L18 Among other differences, there is difference in spatial resolution between OMI and GOME-2 (or SCIAMACHY). Can you comment on how you think this affects the comparison? Can you specify which details of the retrievals might produce the largest differences?

---

## Author Comment (AC1) · 23 Jun 2017

Please, find our response to RC1 in AC1 supplement

Please also note the supplement to this comment:
http://www.atmos-meas-tech-discuss.net/amt-2017-44/amt-2017-44-AC1-supplement.pdf

———————————————————

---

## Author Comment (AC2) · 13 Jul 2017

Response to Referee #4 (doi:10.5194/amt-2017-44-RC2, 2017)

*1) P6 L10-11 How much do you expect this choice of the monthly spectra to affect the correct representation of the day-to-day variability?*

Response to reviewer:
Most OMI trace gas retrieval algorithms, including SPv2.1, use a composite solar spectrum derived from individual irradiance measurements in 2005, in lieu of daily solar observations. Individual daily solar data lack high signal to noise ratio (S/N) important for trace-gas retrievals. Creating monthly averages not only helps improve S/N, but also addresses dynamics of instrumental changes (e.g., wavelength shifts, wavelength-dependent changes in the OMI throughput) that may not be accounted for in the static composite solar spectrum. We have added the following statement in Section 3.1. "The monthly-averaged solar spectra will not capture the daily solar variability, which may differ by about 0.1% around 430 nm and <0.05% elsewhere."

*2) Fig. 3 I think it would be useful to have the same plots for June or July in order to have an idea of the difference at northern high latitudes, which are uncovered in December.*

Response to reviewer:
We agree with reviewer. We updated Fig.3 according to referee #3's suggestion and made a similar figure for July (included as Supplement Figure S1):

[Figure]

Figure S1: OMI NO$_2$ VCDs (top) and difference maps (middle: SPv3 – SPv2) for July 2006: tropospheric VCD (a,c - left) and stratospheric VCD (b,d - right ). Bottom row: change in V$_{trop}$ due to new SCD (bottom - left), and change in V$_{trop}$ due to new *a-priori* NO$_2$ profile shapes (bottom – right).

*3) Fig. 6 This large difference over NCP, China made me think about the impact this change in the algorithm could have on emission estimation, in particular in China where recently significant emission reduction has been probed from space-based observations. Can you estimate or at least speculate on how top-down emission estimates might be affected by these algorithm changes, as compared to some of the conclusions provided in existing literature?*

Response to reviewer:
The impact of algorithm differences on tropospheric $NO_2$ VCDs is more apparent over highly polluted areas such as eastern China in the absolute difference map. However, the relative difference between the two versions is, on average, less than 15% in those areas (Figure R2). Nevertheless, inferred $NO_x$ emissions from the OMI SPv3 tropospheric $NO_2$ VCDs will be lower than those from SPv2.1. Based on prior studies by Lamsal *et al*. (2011), who quantified sensitivity factors of $NO_2$ columns to $NO_x$ emissions ($\beta$) close to unity for highly polluted areas in eastern China, central Europe, and eastern US, we anticipate <15% difference in the inferred top-down NOx emissions. As discussed in this manuscript, these algorithmic changes will have only a minor effect on emissions trends estimates. We have added the following discussion in Section 4.1 in the revised manuscript.
"These changes have a direct implication for derived products, such as the top-down inference of $NO_x$ emissions. Over highly polluted areas, $NO_2$ columns respond nearly linearly to $NO_x$ emissions with a slope close to unity (Lamsal et al., 2011), suggesting that a ~15% lower $V_{trop}$ in SPv3 over eastern China will also be reflected in the inferred $NO_x$ emissions."

[Figure]

Figure R2: Comparison between OMI SPv2 and SPv3 tropospheric NO2 VCDs for July 2006. Left: means (circles) and standard deviations (lines) of SPv2 and SPv3 $V_{trop}$ in intervals of $0.5 \times 10^{15}$ molec cm$^{-2}$. Right: Relative differences (percent) between SPv3 and SPv2 $V_{trop}$ as function of $V_{trop}$.

*4a) Section 5.2 In a previous work (Ialongo et al., 2016) where V2.1 and V3 total columns are compared to Pandora observations in Helsinki (Finland), the difference between V3 and V2.1 retrievals are quite systematic, with V3 sensibly smaller than Pandora (as compared to V2.1). Can you speculate about the difference between your results with this previous paper?*

Response to reviewer:

Comparison between low spatial resolution satellite $NO_2$ data, such as OMI, and point ground based measurements, such as Pandora, depends strongly on pollution level and the spatial inhomogeneity of the $NO_2$ field, local $NO_2$ emission sources, $NO_2$ vertical distribution, type of terrain, solar elevation, aerosol pollution, that vary greatly from place to place. Hong Kong is unique in that new OMI SPv3 data are close to the previous version (Fig. 8). This could be due to the opposite effects of smaller SCDs and smaller AMFs due to the higher spatial resolution of the *a priori* $NO_2$ profile shapes (Fig. 3). For most other polluted locations the new SPv3 data are lower that the previous version, as confirmed in Pandora comparisons in Helsinki (Ialongo et al., 2016). The detailed validation of the OMI $NO_2$ product, which addresses all relevant factors, will be conducted in a follow up paper.

We have added the following sentences at the end of Sec 5.2 and a new reference:

"Hong Kong is unique in that new OMI SPv3 data are close to the previous version (*cf.* the bottom row of panels Fig. 8). This could be due to the opposite effects of smaller SCDs and smaller AMFs due to the higher spatial resolution of the *a priori* $NO_2$ profile shapes (Figure 3). For most other polluted locations the new SPv3 data are lower than the previous version, as confirmed with direct-sun Pandora comparisons in Helsinki (Ialongo et al., 2016)."

*4b) In general, I think the manuscript could get stronger with slightly more comparison to independent ground-based observations, similarly to what was presented by (Lamsal et al., 2014). While the main scope of this manuscript might not be the comprehensive validation of the V3 product (this could be addressed in a separate paper, perhaps), I would suggest including one or two more pictures, for example something similar to fig. 6 and 7 in Lamsal et al. (2014), including for example Pandora measurements or additional max-doas stations. According to this previous paper, such ground-based data should be available to the authors.*

Response to reviewer:
Our on-going validation work will be presented in a separate paper. In this paper, we focus on changes from the earlier version and present only initial consistency checks with other data sets. Sparse and short-term ground $NO_2$ measurements (*e.g.*, MAX-DOAS, Pandora), incomplete information (*e.g.*, aircraft and surface network data), preferential placement of ground-based instruments, and the need for assessing the validation data themselves makes validation of OMI $NO_2$ challenging. For instance, most Pandora measurements have been conducted in polluted urban areas with strong $NO_2$ spatial gradients, and data records are not yet long

enough to help satellite validation definitively. Challenges remain to fully understand the discrepancies observed even at clean sites, such as Mauna Loa, as illustrated in Figure R3. This figure shows a comparison of coincident OMI and Pandora total $NO_2$ VCDs at the Mauna Loa Observatory site on the Island of Hawaii. Because of its elevation (~3km), the site allows for the measurement of mostly free-tropospheric and stratospheric $NO_2$ columns. We compared Pandora observations during OMI overpasses (+/-10 minutes) for mostly clear-sky days (OMI cloud radiance fraction < 0.6) in 2014-2016. These comparisons show fairly low correlation (r = 0.3, N = 175) and a low bias (SPv2 or SPv3 OMI<Pandora) inconsistent with FTIR intercomparison results at Tenerife (Fig. 9). Also, the seasonal summer $NO_2$ enhancement measured by Pandora is not confirmed by OMI observations or GMI simulation. This preliminary comparison warrants further assessment and evaluation of the Pandora retrievals themselves. A detailed validation is an undertaking that involves a more thorough investigation and is beyond the scope of this paper. We have added the following paragraph in the revised manuscript at the beginning of section 5.

"We assess OMI SPv3 data by comparing with other independent observations. Here we present only initial consistency checks with other data sets. Sparse and short-term ground-based NO2 measurements, incomplete information, preferential placement of ground-based instruments, and the need for assessing the validation data themselves make validation of satellite NO2 retrievals challenging and warrant detailed validation work."

[Figure]

Figure R3. OMI, GMI and Pandora total $NO_2$ monthly mean VCDs at the clean mountain Mauna Loa site on the Island of Hawaii in 2014-2016.

*5) Vasilkov et al. (2017) discussed the effect of the varying observation geometry on the NO2 vertical column retrieval. Their findings suggest that the NO2 vertical columns would typically increase (at least in this test orbit over the American continent), when taking into account a geometry-dependent Lambertian equivalent reflectivity (LER) in the NO2 retrieval algorithm. Can you comment in the manuscript on how this could affect your retrievals?*

Response to reviewer:
We agree that applying new MODIS-based geometry dependent surface reflectance would typically decrease tropospheric AMFs and increase tropospheric VCDs for polluted sites under clear sky conditions. However, this effect would be practically negligible for cloudy conditions and for the free tropospheric and stratospheric $NO_2$ VCDs.

We have added the following statement in our conclusions:
"However, applying a new geometry dependent Lambertian equivalent reflectivity in AMF calculation would result in increasing tropospheric VCDs (Vasilkov et al., 2017) as well as derived top-down $NO_x$ emissions and surface concentrations."

*6) Section 5.3 P14 L18 Among other differences, there is difference in spatial resolution between OMI and GOME-2 (or SCIAMACHY). Can you comment on how you think this affects the comparison? Can you specify which details of the retrievals might produce the largest differences?*

Response to reviewer:
This comparison over the Pacific region is unlikely to be affected significantly by the different spatial resolutions of the various satellite instruments. The stratospheric $NO_2$ field is reasonably homogenous on these spatial scales (Fig. 11b), as are the total VCDs in the absence of finer-scale spatial gradients associated with tropospheric pollution sources (Fig. 11a).

Even though the measurements were not truly coincident, these independent instruments and retrievals are in agreement to within their uncertainties. Attribution of differences to specifics of the retrievals themselves is therefore challenging. One way to approach the question of retrieval-dependent differences would be to apply the OMI retrieval to the other datasets, to the extent possible, and evaluate how the retrieved quantities change.